# A non-canonical role for the EDC4 decapping factor in regulating MARF1-mediated mRNA decay

William R Brothers[1], Steven Hebert[1], Claudia L Kleinman[1,2], Marc R Fabian[1,3,4]*

[1]Lady Davis Institute for Medical Research, Jewish General Hospital, Montreal, Canada; [2]Department of Human Genetics, McGill University, Montreal, Canada; [3]Department of Biochemistry, McGill University, Montreal, Canada; [4]Department of Oncology, McGill University, Montreal, Canada

**Abstract** EDC4 is a core component of processing (P)-bodies that binds the DCP2 decapping enzyme and stimulates mRNA decay. EDC4 also interacts with mammalian MARF1, a recently identified endoribonuclease that promotes oogenesis and contains a number of RNA binding domains, including two RRMs and multiple LOTUS domains. How EDC4 regulates MARF1 action and the identity of MARF1 target mRNAs is not known. Our transcriptome-wide analysis identifies bona fide MARF1 target mRNAs and indicates that MARF1 predominantly binds their 3' UTRs via its LOTUS domains to promote their decay. We also show that a MARF1 RRM plays an essential role in enhancing its endonuclease activity. Importantly, we establish that EDC4 impairs MARF1 activity by preventing its LOTUS domains from binding target mRNAs. Thus, EDC4 not only serves as an enhancer of mRNA turnover that binds DCP2, but also as a repressor that binds MARF1 to prevent the decay of MARF1 target mRNAs.

## Introduction

The regulated decay of eukaryotic mRNA populations plays an important role in the post-transcriptional control (PTC) of gene expression. These PTC programs are, in turn, critical for regulating a number of biological processes, including during early development, cell proliferation and immune response. The major mRNA decay pathway in eukaryotes initiates with the shortening of the mRNA poly(A) tail (deadenylation), which is carried out by the CCR4-NOT deadenylase complex (*Yamashita et al., 2005*). mRNA deadenylation is then followed by recruitment of the DCP1-DCP2 decapping complex that hydrolyzes the mRNA 5'-cap structure. This complex is comprised of the DCP2 decapping enzyme, DCP1 (associates with DCP2) and enhancers of mRNA decapping (EDC) proteins (i.e. EDC3 and EDC4). EDC4 enhances mRNA decapping by binding both DCP1 and DCP2 stimulating DCP2 activity (*Chang et al., 2014*). Once the 5'-cap is removed to yield a 5' monophosphate, the mRNA is committed for degradation by the 5'-to-3' exonuclease XRN1. Many proteins that play a role in mRNA metabolism, including EDC4 and other mRNA decapping factors, are found associated with processing (P)-bodies, cytoplasmic ribonucleoprotein granules that contain translationally repressed mRNAs (*Hubstenberger et al., 2017*; *Standart and Weil, 2018*). In addition to binding DCP1 and DCP2, human EDC4 also interacts with the RNA binding protein MARF1 (meiosis regulator and mRNA stability factor 1), a recently characterized endonuclease that promotes oogenesis (*Bloch et al., 2014*; *Kanemitsu et al., 2017*; *Nishimura et al., 2018*; *Su et al., 2012a*; *Yao et al., 2018*).

Mammalian MARF1 proteins contain a NYN endonuclease domain (*Figure 1A*) that adopts a PIN-like fold (*Nishimura et al., 2018*; *Su et al., 2012a*; *Yao et al., 2018*). In addition to its NYN domain, MARF1 also contains a number of RNA binding modules, including two RRMs and eight tandem

*For correspondence:
marc.fabian@mcgill.ca

Competing interests: The authors declare that no competing interests exist.

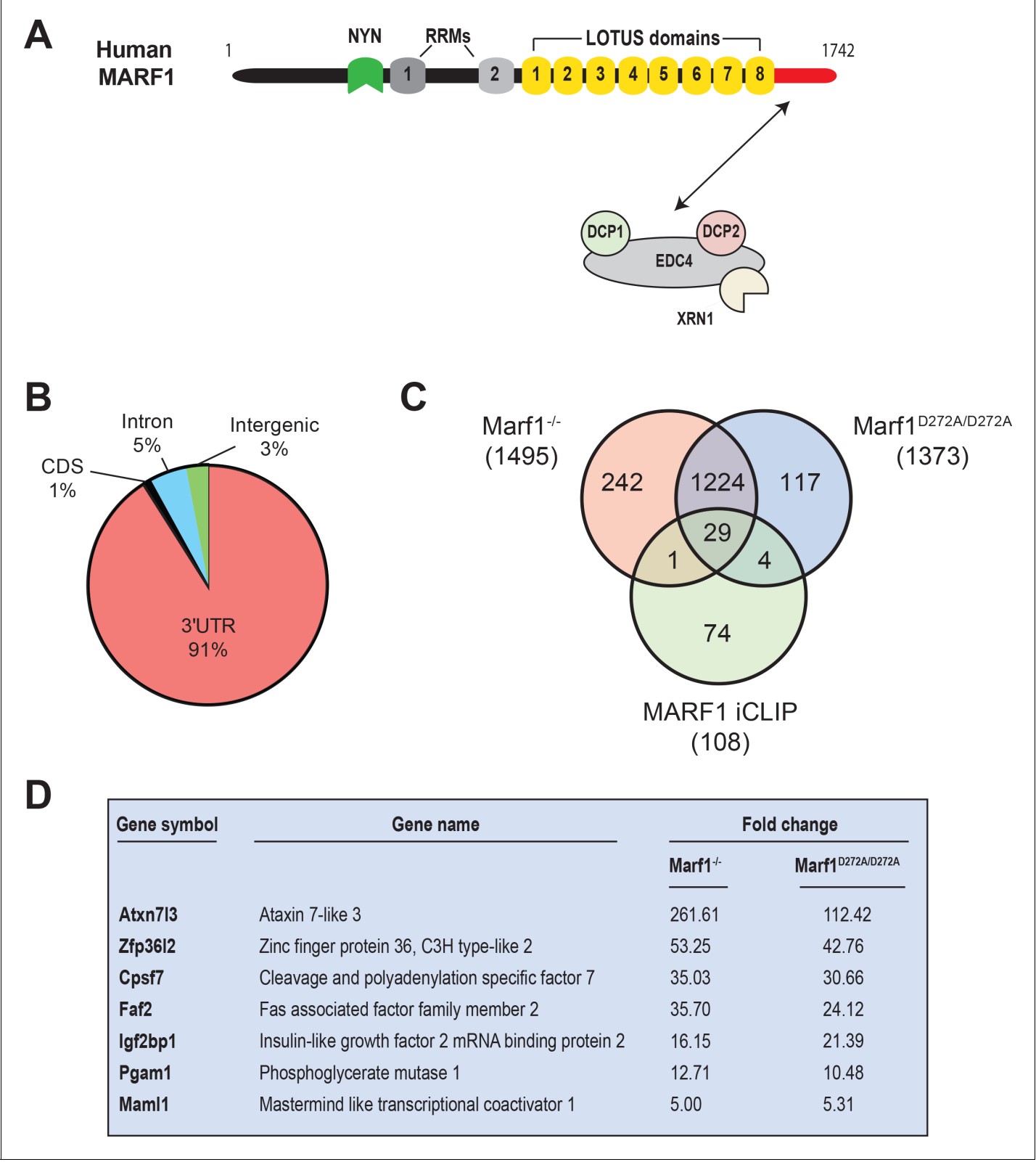

**Figure 1.** Identification of human MARF1 target mRNAs. (**A**) Schematic diagram of full-length MARF1. (**B**) Distribution of crosslinked sequence reads. (**C**) Venn diagram illustrating the relationship of MARF1 target mRNAs identified by iCLIP in HEK293 cells with transcripts that were upregulated in *Marf1*[-/-] and *Marf1*[D272A/D272A] and germinal vesicle-stage mouse oocytes as compared to wild-type. (**D**) A partial list of mRNAs identified by iCLIP that were upregulated in both *Marf1*[-/-] and *Marf1*[D272A/D272A] and germinal vesicle-stage mouse oocytes as compared to wild-type.

*Figure 1 continued on next page*

*Figure 1 continued*

The online version of this article includes the following figure supplement(s) for figure 1:

**Figure supplement 1.** Representative FLAG-MARF1^ΔNYN iCLIP coverage at the loci of ATXN7L3, IGFP2BP1, MAML1, FAF2 and PGAM2 loci shows specificity for the 3'UTR.

winged helix-turn-helix (wHTH) LOTUS (Limkain, Oskar, and Tudor containing proteins 5 and 7) domains. *Drosophila* MARF1 also contains several tandem LOTUS domains, but has only one RRM and lacks a nuclease domain (*Zhu et al., 2018*). Instead, the first of its LOTUS domains recruits the CCR4-NOT complex to initiate deadenylation-dependent mRNA decay. In contrast, mammalian MARF1 does not directly associate with the CCR4-NOT complex but rather uses a C-terminal motif to physically interact with EDC4, DCP1 and DCP2 (*Bloch et al., 2014*; *Nishimura et al., 2018*). How EDC4 and other decapping factors regulate MARF1 action is not known.

MARF1 is robustly expressed in mouse oocytes where it plays a critical role in regulating their meiotic progression based on the finding that female *Marf1^-/-* mice are sterile (*Su et al., 2012a*). In addition to regulating the development of the mammalian germline, MARF1 is also expressed in the developing brain where it has been reported to regulate neuronal differentiation in the embryonic cortex (*Kanemitsu et al., 2017*). While knocking out MARF1 in oocytes dramatically alters gene expression, it is currently unclear which mRNAs are directly targeted by MARF1 and which of its RNA binding modules mediate target RNA recognition.

To investigate which mRNAs are directly targeted by MARF1 and how MARF1 interfaces with them, we carried out transcriptome-wide analysis of MARF1-targeted mRNAs by individual-nucleo-tide resolution UV crosslinking and immunoprecipitation (iCLIP). We demonstrate that MARF1 inter-acts with a select set of mRNAs by predominantly binding to their 3'UTRs. We further show that MARF1 utilizes its tandem LOTUS domains to bind target mRNAs, with several core LOTUS domain being essential. While the MARF1 LOTUS domains are involved in target recognition, we further show that RRM1 plays a critical role in NYN-mediated decay of targets following initial MARF1 bind-ing. Importantly, we demonstrate that EDC4 binding to MARF1 impairs MARF1-mediated repression by preventing MARF1 from binding to target mRNAs.

## Results

### Human MARF1 protein binds to the 3'UTR of target mRNAs

To comprehensively identify MARF1-associated mRNAs, we performed iCLIP using engineered HEK293 cells stably expressing a doxycycline (Dox)-inducible FLAG-tagged MARF1 that lacks RNAse activity (F-MARF1^ΔNYN). Briefly, F-MARF1^ΔNYN -expressing cells were UV-crosslinked, cell lysates were partially digested with RNAseI and MARF1-RNA complexes were subsequently immunoprecipitated with a FLAG antibody. RNA fragments bound to MARF1 were then isolated, converted into cDNA libraries and analyzed by deep sequencing. We also carried out a parallel iCLIP experiment with a FLAG antibody using control HEK293 cells that do not express a FLAG-tagged MARF1 protein. This allowed us to stringently control for non-specific background in FLAG immunoprecipitations. Recovered RNAs from two biological experiments were sequenced, PCR artifacts and multi-mapping reads were removed and primary genome-aligned reads were clustered to generate peaks (*Supplementary file 1*). This analysis identified only 108 high-confi-dence mRNAs bound by F-MARF1^ΔNYN with the vast majority of assigned peaks mapping to 3'UTRs (*Figure 1B* and exemplified in *Figure 1—figure supplement 1*). The observation that most of the crosslinked reads derived from exonic sequences is consistent with the cytosolic localization of MARF1 (*Bloch et al., 2014*).

It was recently reported that disruption of the *Marf1* gene by genetrap (GT) or inserting an inacti-vating mutation into the MARF1 NYN domain (D272A) led to dramatic changes in oocyte gene expression (*Yao et al., 2018*). A comparison of our iCLIP data with these gene expression datasets identified 34 iCLIP targets that were upregulated in *Marf1^GT/GT* or *Marf1^D272A/D272A* oocytes, with the majority of these targets (29) being upregulated in both contexts (*Figure 1C* and *Supplementary file 2*). These include *Maml1*, *Zfp36l2*, *Atxn7l3*, *Faf2*, *Pgam1*, *Igf2bp1* and *Cpsf7*, whose levels were increased anywhere from 5- to 261-fold (*Figure 1D*).

Full-length λNHA-MARF1 efficiently represses a *Renilla* luciferase (RL) reporter mRNA in HEK293 cells when it is artificially tethered via the bacteriophage λN-BoxB tethering system to its 3'UTR (RL-5BoxB) (*Figure 2A* through C) (*Bos et al., 2016*; *Nishimura et al., 2018*). Moreover, this repression is NYN-dependent as a MARF1 mutant lacking the NYN domain (λNHA-MARF1$^{\Delta NYN}$) did not efficiently silence the RL-5BoxB reporter. To determine if MARF1 has the potential to repress targets identified by iCLIP, we constructed a set of RL reporters containing 3'UTRs of MARF1 targets, including those of *Maml1*, *Notch2*, *Zfp36l2*, *Atxn7l3*, *Faf2*, *Pgam1*, *Igf2bp1* and *Cpsf7*. We also generated a RL reporter with the 3'UTR of Interleukin 6 (IL-6), an mRNA 3'UTR that was not identified as a MARF1 target. MARF1-expressing plasmids were transfected along with RL-3'UTR constructs and a FL construct as a transfection control. In keeping with our results using the RL-5BoxB reporter, full-length λNHA-MARF1 efficiently repressed these reporters in a NYN-dependent manner (*Figure 2D*). Moreover, repression was specific to *bona fide* targets, as MARF1 was not able to silence the IL-6 reporter. This repression also appeared to be at the level of mRNA decay, as full-length MARF1 significantly reduced the steady state levels of RL-MAML1 and RL-NOTCH2 mRNAs (*Figure 2E*) and the stability of RL-MAML1 mRNA (*Figure 2F*) in a NYN-dependent manner, as assessed by RT-qPCR. Similarly, overexpressing wild-type MARF1 resulted in marked decrease of the steady-state levels of both endogenous *Notch2* and *Maml1* mRNAs as compared to cells expressing MARF1$^{\Delta NYN}$ (*Figure 2G*). Although our iCLIP data demonstrated MARF1 binding to numerous mRNA 3'UTRs, we were not able to identify any short sequence motifs enriched in MARF1 peaks. This suggests that structural motifs in the target RNAs and/or other positional elements are likely to determine the 3'UTR binding specificity of MARF1.

## MARF1 recognizes endogenous targets using a core subset of LOTUS domains

MARF1 contains several domains with RNA binding capacity, including two RRMs and eight LOTUS domains as well as a C-terminal motif that interacts with mRNA decapping factors (*Nishimura et al., 2018*). We set out to determine how MARF1 recognizes target mRNAs by generating a number of C-terminal deletion mutants (*Figure 3A*). A MARF1 mutant lacking the decapping factor interaction motif (ΔC-term) efficiently silenced both the RL-MAML1 and the RL-5BoxB reporters (*Figure 3B*). In contrast, a MARF1 N-terminal fragment (λNHA-MARF1$^{N-term}$) that lacks both the LOTUS domains and the decapping factor interaction motif efficiently silenced RL-5BoxB but failed to repress the RL-MAML1 reporter. Sequence alignment of human MARF1 protein LOTUS domains revealed that the central LOTUS domains 3 and 5 are highly conserved in MARF1 homologs, including *Xenopus tropicalis*, *Drosophila melanogaster* and *Dario rerio* (*Figure 3—figure supplement 1A*). In addition, human MARF1 LOTUS domains 3 and 5 also share a high degree of sequence identity with each other (*Figure 3—figure supplement 1B*). Based on these data, we hypothesized that specific LOTUS domains may play a more important role in MARF1-mediated repression than others. To determine which LOTUS domains in MARF1 are required for target RNA repression, we deleted each LOTUS domain individually from full-length λNHA-MARF1 and tested their abilities to repress RL-MAML1 (*Figure 3C* and *Figure 3—figure supplement 1*). We observed that MARF1 mutants lacking LOTUS domains 1, 2, 6, 7 or eight efficiently silenced RL-MAML1 and RL-5BoxB mRNAs. However, MARF1 mutants lacking LOTUS domains 3, 4 or 5 were unable to efficiently silence RL-MAML1. Nevertheless, MARF1 proteins lacking LOTUS domains 3, 4 or 5 efficiently repressed the RL-5BoxB reporter mRNA, suggesting that deleting these core LOTUS domains did not impact MARF1 nuclease activity but rather its binding to endogenously targeted mRNA 3'UTRs (*Figure 3C*). In addition to not being able to repress RL-MAML1, we also observed that λNHA-MARF1 lacking the LOTUS domain 4 (λNHA-MARF1$^{\Delta LOTUS4}$) failed to silence a number of other RL reporters containing MARF1 target 3'UTRs, including *Zfp36l2*, *Atxn7l3*, *Faf2*, *Igf2bp1* and *Cpsf7* (*Figure 3D*). All in all, these data indicated that MARF1 utilizes its central LOTUS domains to bind target mRNAs.

## MARF1 RRM1 is critical for NYN activity

To examine if MARF1 also utilizes its RRMs to bind targeted mRNAs we also generated a series of MARF1 deletion mutants lacking either RRM1 (λNHA-MARF1$^{\Delta RRM1}$), RRM2 (λNHA-MARF1$^{\Delta RRM2}$), or both RRMs (λNHA-MARF1$^{\Delta RRMs}$) (*Figure 4A*). MARF1-expressing plasmids were transfected along with RL-MAML1 target mRNA plasmid and a FL reporter plasmid as a

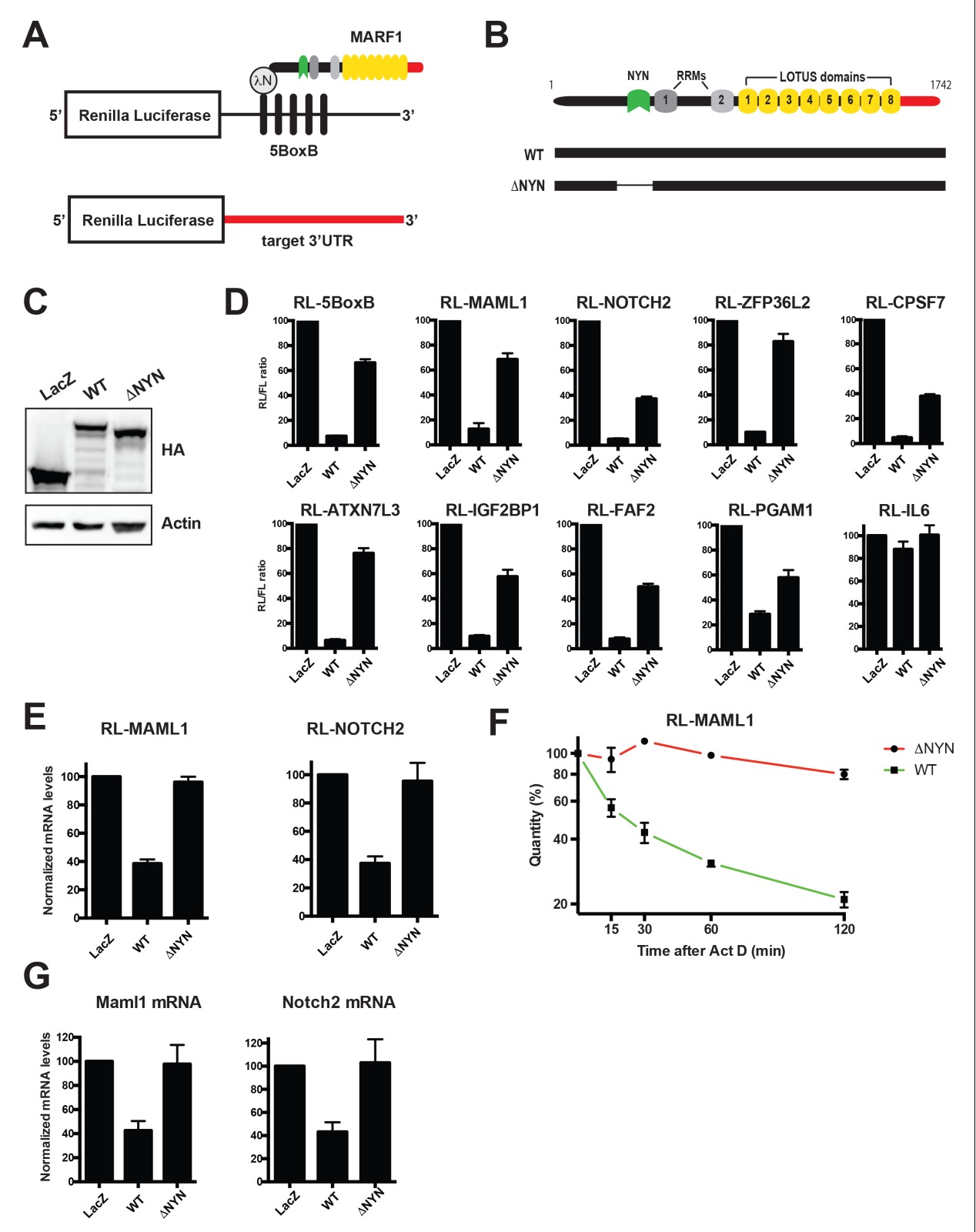

**Figure 2.** MARF1 represses target mRNAs via their 3'UTRs. (**A**) Schematic diagram of the basic *Renilla* luciferase (RL)-encoding mRNA reporter, containing five 19-nt BoxB hairpins, interacting with λN-HA-MARF1, as well as a RL reporter with a 3'UTR from endogenous mRNAs. (**B**) Schematic diagram of full-length MARF1 and MARF1 fragments used in tethering assays. (**C**) Western blot analysis of HEK293 cells expressing indicated proteins. (**D**) RL activity detected in extracts from HEK293 cells expressing the indicated proteins. Cells were cotransfected with constructs expressing the RL-

*Figure 2 continued on next page*

*Figure 2 continued*

5BoxB reporter, FL, and indicated fusion proteins. Histograms represented normalized mean values of RL activity from a minimum of three experiments. RL activity values seen in the presence of λNHA-LacZ were set as 100. (E) RL-MAML1 (left panel) and RL-NOTCH2 (right panel) mRNA levels detected in extracts from HEK293 cells expressing the indicated proteins. Histograms represented mean values of RL-MAML1 or RL-NOTCH2 mRNAs normalized to FL mRNA from a minimum of three experiments. mRNA levels values seen in the presence of λNHA-LacZ were set as 100. (F) The stability of RL-MAML1 was assessed by using actinomycin D (5 µg/ml) for the indicated amount of time. Total RNA was isolated, reverse transcribed and RL-MAML1 RNA was quantified by qPCR. RL-MAML1 mRNA decay rates were normalized to FL mRNA levels with the zero time point set at 100. (G) Endogenous MAML1 and NOTCH2 mRNA levels detected in extracts from HEK293 cells expressing the indicated proteins. mRNA levels were normalized to GAPDH mRNA levels for a minimum of three experiments. mRNA levels in the presence of λNHA-LacZ were set as 100. Error bars represent the SEM of multiple independent experiments.

transfection control (*Figure 4A and B*). We observed that deleting both RRMs rendered MARF1 unable to repress RL-MAML1. Moreover, while λNHA-MARF1$^{\Delta RRM2}$ efficiently silenced the RL-MAML1 reporter, λNHA-MARF1$^{\Delta RRM1}$ was unable to do so. Similarly, λNHA-MARF1$^{\Delta RRM1}$ was unable to efficiently repress other RL reporter mRNAs containing MARF1-targeted 3'UTRs, including ATXN7L3, CPSF7, IGF2BP1, NOTCH2, PGAM1 and ZFP36L2 (*Figure 4—figure supplement 1A*). Previous studies established that RRM1 binds single-stranded RNA (*Yao et al., 2018*). To assess if the RNA-binding capacity of RRM1 plays a role in MARF1-mediated repression we mutated a number of amino acid residues on the predicted canonical RNA-binding surface of the RRM1 domain (Y515A, Y517A, I552A, F582A), which would be expected to abolish its RNA binding capacity (λNHA-MARF1$^{RRM1mut}$) (*Figure 4—figure supplement 1B*). In agreement with this hypothesis, this RRM1 mutant was unable to silence the RL-MAML1 reporter (*Figure 4C*). In addition to using reporter mRNAs containing MARF1-targeted 3'UTRs, we also tested the silencing capacity of these λN-tagged MARF1 mutant constructs by tethering them to RL-5BoxB mRNA to rule out any mutations interfering with MARF1 function independent of its mode of recruitment (*Figure 4D*). Surprisingly, deleting or mutating RRM1 also abrogated λNHA-MARF1 repression of the RL-5BoxB reporter RNA. In addition, deleting RRM1 in the context of λNHA-MARF1$^{N-term}$ also abolished the repressive activity on RL-5BoxB mRNA (*Figure 4E and F*). These results are in contrast to deleting the MARF1 LOTUS domains, which impaired only reporters with endogenous 3'UTRs and had no effect on the RL-5BoxB mRNA (*Figure 3*). Taken together, these data indicate that RRM1 is required for NYN-mediated cleavage of MARF1-targeted mRNAs irrespective of how MARF1 is recruited to the targeted mRNA.

## The mRNA decapping factor EDC4 antagonizes MARF1 repression

EDC4 serves as a large scaffold that binds both DCP1 and DCP2 thereby promoting their association and mRNA decapping (*Chang et al., 2014*; *Yu et al., 2005*). In addition, EDC4 has been reported to associate with miRISC as well as the RNA binding protein TTP and enhance miRNA-mediated gene silencing (*Brodersen et al., 2008*; *Eulalio et al., 2007*; *Nishihara et al., 2013*). MARF1 also physically interacts via its C-terminal motif with EDC4 (*Bloch et al., 2014*; *Fenger-Grøn et al., 2005*; *Hubstenberger et al., 2017*; *Nishimura et al., 2018*; *Youn et al., 2018*). Nevertheless, a MARF1 mutant lacking its C-terminal motif (λNHA-MARF1$^{\Delta C-term}$) showed no functional defect in repressing a RL-5BoxB reporter construct when artificially tethered to it or the RL-MAML1 reporter (*Figure 3B*). We speculated that endogenous EDC4 levels may be limiting in cells such that the majority of ectopic MARF1 is not bound by endogenous decapping factors to enhance repression of target mRNAs. To test this hypothesis, we co-transfected the RL-MAML1 reporter with decreasing amounts of plasmid encoding either full-length λNHA-MARF1 or λNHA-MARF1$^{\Delta C-term}$ to titrate MARF1 expression (*Figure 5A*). Surprisingly, we observed that at low levels of expression, full-length MARF1 did not repress the RL-MAML1 reporter, whereas λNHA-MARF1$^{\Delta C-term}$ remained able to silence RL-MAML1 mRNA when expressed at a comparable level (*Figure 5B*). In addition, this effect was specific to how MARF1 was recruited to a target mRNA, as low-level expression of λNHA-MARF1 efficiently repressed RL-5BoxB when artificially tethered (*Figure 5C*). Based on these data, we hypothesized that MARF1 association with EDC4 and other decapping factors may impair its ability to repress a target mRNA rather than enhance it. To determine if decapping factors play a role in impairing MARF1 function when expressed at low levels we depleted endogenous EDC4 and DCP2 by RNA interference. Knocking down EDC4 rescued wild-type λNHA-MARF1-mediated repression

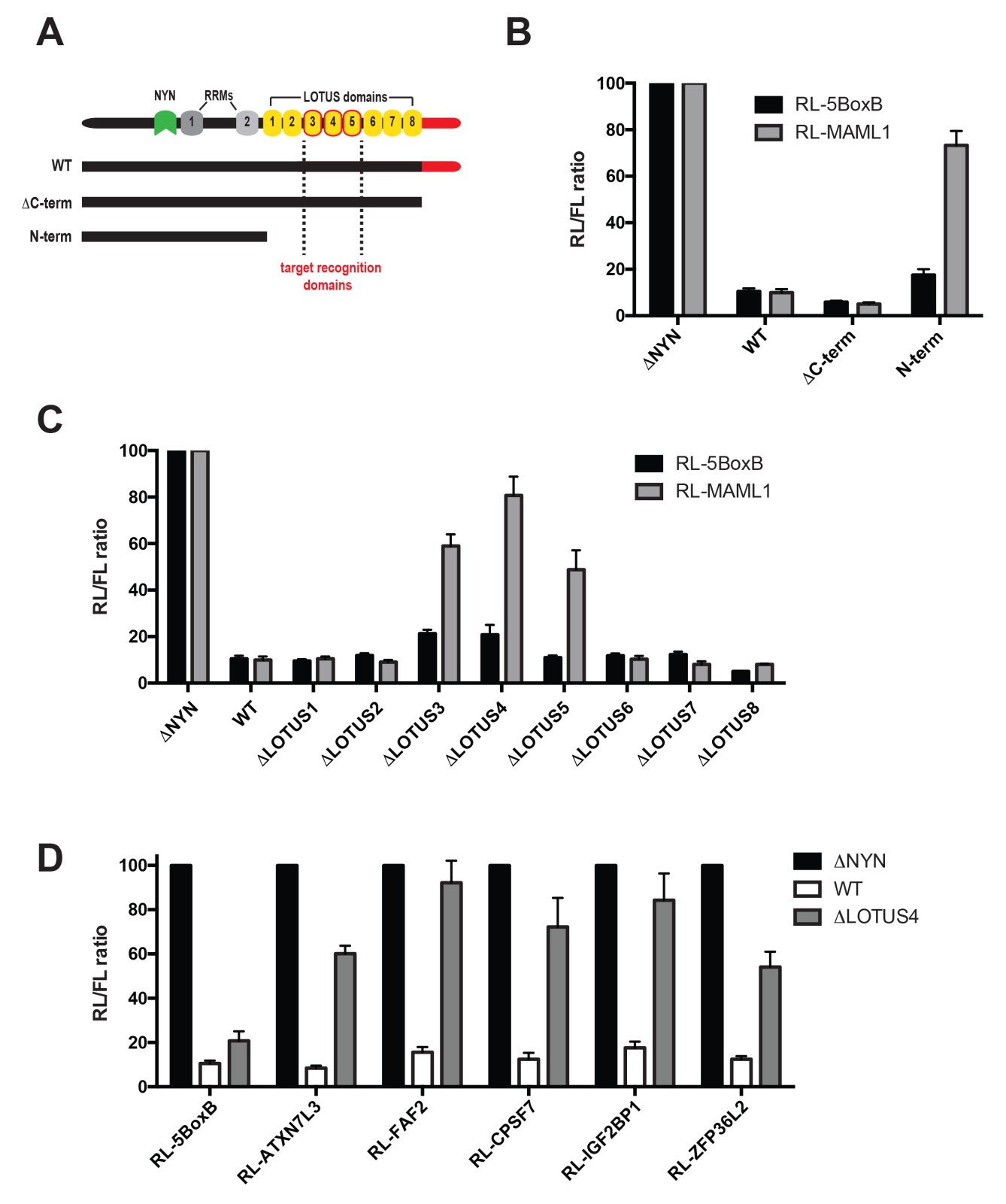

**Figure 3.** Central MARF1 LOTUS domains are required to silence mRNAs containing 3'UTRs of MARF1 target mRNAs. (**A**) Schematic diagram of wild-type MARF1 protein, as well C-terminal deletion mutants. Core LOTUS domains required for MARF1-mediated repression are bordered in red. (**B–D**) RL activity detected in extracts from HEK293 cells expressing the indicated proteins. Cells were cotransfected with constructs expressing the RL-MAML1

*Figure 3 continued on next page*

*Figure 3 continued*

reporter or RL-5BoxB reporter, along with FL, and indicated fusion proteins. Histograms represented normalized mean values of RL activity from a minimum of three experiments. RL activity values seen in the presence of λNHA-MARF1$^{\Delta NYN}$ were set as 100.

The online version of this article includes the following source data and figure supplement(s) for figure 3:

**Source data 1.** Comparative sequence analysis of MARF1 LOTUS domains.

**Figure supplement 1.** Western blot analysis of HEK293 cells expressing λNHA-MARF1 mutants lacking the indicated LOTUS domains.

to the level achieved by λNHA-MARF1$^{\Delta C\text{-term}}$ (*Figure 5D and E*). EDC4 depletion also co-depleted DCP2 (*Figure 5D*), as has been previously reported (*Erickson et al., 2015*). However, knocking down DCP2, which does not change EDC4 expression (*Figure 5D*), did not enhance λNHA-MARF1 repression of RL-MAML1 (*Figure 5E*).

To test if overexpressing EDC4 could impact MARF1-mediated repression when MARF1 is expressed at high levels we co-transfected cells with a plasmid coding for V5-tagged EDC4 along with expression plasmids for either the RL-MAML1 or RL-NOTCH2 reporters and high levels of full-length λNHA-MARF1 or λNHA-MARF1$^{\Delta C\text{-term}}$ (*Figure 5F and G* and *Figure 5—figure supplement 1*). Overexpressing EDC4 dramatically impaired wild-type λNHA-MARF1-mediated repression of RL-MAML1 and RL-NOTCH2 reporters, as compared to cells expressing only endogenous EDC4. In contrast, EDC4 overexpression had no effect on the repressive activity of λNHA-MARF1$^{\Delta C\text{-term}}$. Moreover, EDC4 had no noticeable impact on the ability of full-length λNHA-MARF1 to repress a RL-5BoxB mRNA (*Figure 5H*). To demonstrate that the inhibitory effect of EDC4 on MARF1-mediated gene repression is not unique to the luciferase reporter system, the steady state mRNA levels of *Maml1* and *Notch2* were assessed by RT-qPCR in cells transfected with either λNHA-LacZ, λNHA-MARF1, λNHA-MARF1$^{\Delta NYN}$, or λNHA-MARF1$^{\Delta C\text{-term}}$ and co-transfected with V5-tagged EDC4 (*Figure 5I and J*). Consistent with the luciferase reporter data, overexpression of EDC4 increased the steady state levels of *Maml1* and *Notch2* mRNAs in λNHA-MARF1 expressing cells but had no effect on their levels in λNHA-MARF1$^{\Delta C\text{-term}}$ expressing cells. Taken together, these data indicate that EDC4 antagonizes MARF1-mediated repression of target mRNAs.

## EDC4-MARF1 interaction localizes MARF1 to P-bodies and impairs MARF1-mediated repression

MARF1 has been reported to localize with EDC4 in P-bodies (*Bloch et al., 2014*; *Hubstenberger et al., 2017*; *Youn et al., 2018*). We next set out to determine if MARF1 utilizes its C-terminal domain, which interacts with EDC4 (*Figure 6A*), to localize to P-bodies and whether disrupting MARF1 localization to P-bodies enhances its ability to repress a target mRNA. To this end, we used immunofluorescence microscopy to assess the subcellular localization of our HA-tagged wild-type and C-terminal deletion mutant MARF1 proteins in HeLa cells expressing V5-tagged EDC4. Full-length λNHA-MARF1 localized to P-bodies, as determined by co-localization with EDC4 (*Figure 6B*, left panels), whereas a λNHA-MARF1 mutant lacking its C-terminal motif did not localize with EDC4 to P-bodies but rather exhibited a diffuse cytoplasmic distribution (*Figure 6B*, middle panels). To test whether the interaction between EDC4 and MARF1 is what is responsible for EDC4 impairing the repressive activity of MARF1, we took advantage of the MARF1 C-terminal motif, which is sufficient on its own to interact with EDC4 (*Nishimura et al., 2018*) and also co-localizes with EDC4 in P-bodies (*Figure 6—figure supplement 1A and B*). We hypothesized that expressing MARF1$^{C\text{-term}}$ might rescue wild-type MARF1 silencing in trans by competing for binding to EDC4, thus preventing EDC4 from binding to wild-type MARF1 (*Figure 6A*). To this end, wild-type λNHA-MARF1 or the C-terminal deletion mutant were expressed at low levels, along with the RL-MAML1 reporter and a plasmid encoding a FLAG-tagged MARF1$^{C\text{-term}}$ fragment. Similar to the effects observed upon EDC4 depletion, expressing the MARF1$^{C\text{-term}}$ enhanced wild-type MARF1-mediated silencing of the RL-MAML1 reporter to the level of repression observed with the λNHA-MARF1$^{\Delta C\text{-term}}$ protein (*Figure 6C*). In addition, expression of FLAG-tagged MARF1$^{C\text{-term}}$ shifted the distribution of wild-type MARF1 from strongly co-localizing with EDC4 in P-bodies to being more diffuse in the cytoplasm (*Figure 6B*, right panels). Taken together, these data indicate that the EDC4-MARF1 interaction localizes MARF1 to P-bodies and prevents MARF1 from efficiently repressing target mRNAs.

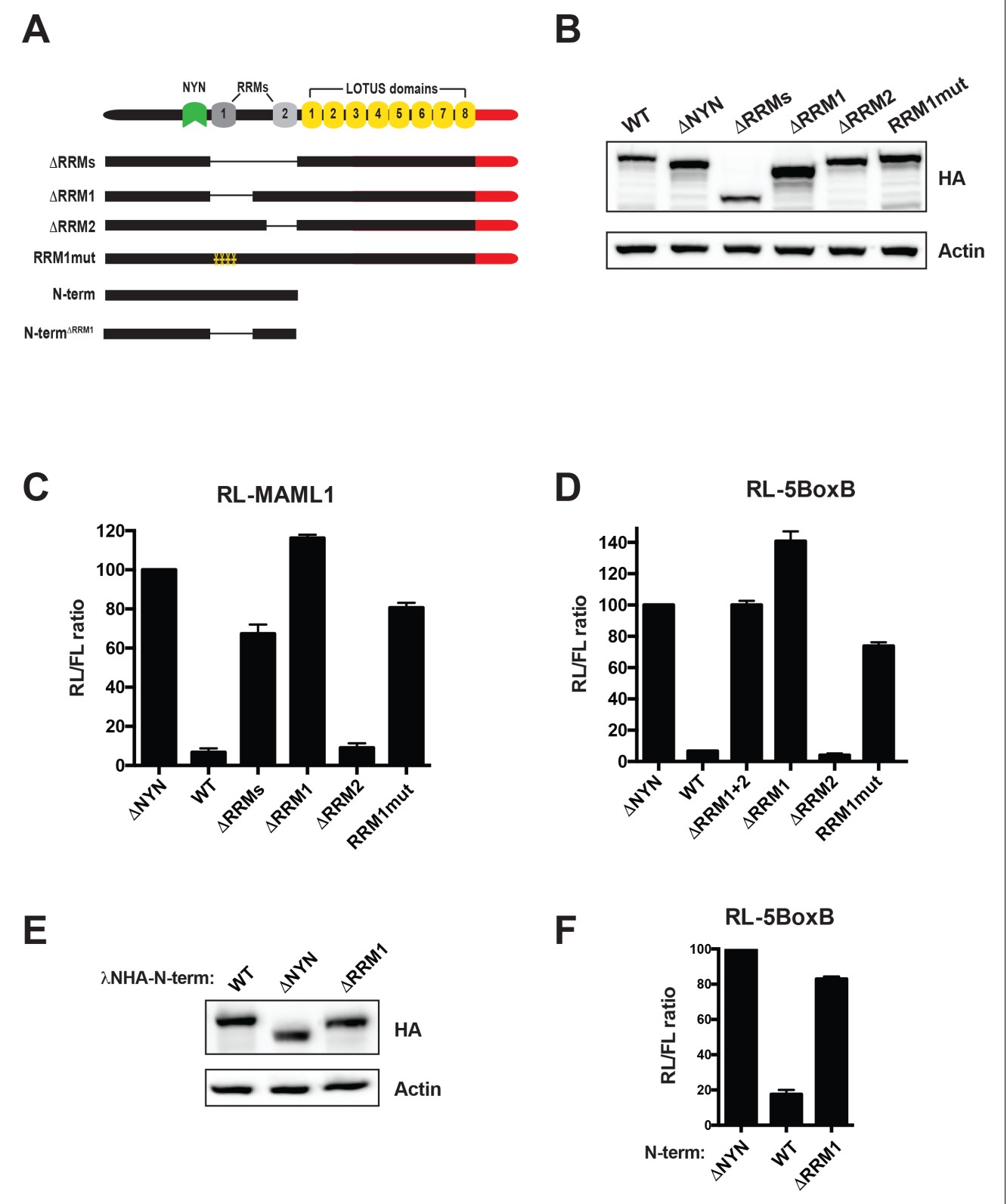

**Figure 4.** MARF1 RRM1 is required to silence target mRNAs. (**A**) Schematic diagram of wild-type MARF1 protein, as well as RRM mutants. (**B and E**) Western blot analysis HEK293 cells transfected with plasmids expressing full-length (**B**) or N-terminal fragments (**E**). (**C, D and F**) RL activity detected in extracts from HEK293 cells expressing the indicated proteins. Cells were cotransfected with constructs expressing the RL-MAML1 reporter (**C**) or the RL-

*Figure 4 continued on next page*

*Figure 4 continued*

5BoxB reporter (**D and F**), FL, and indicated fusion proteins. Histograms represented normalized mean values of RL activity from a minimum of three experiments. RL activity values seen in the presence of λNHA-MARF1$^{\Delta NYN}$ were set as 100. (**E**).

The online version of this article includes the following figure supplement(s) for figure 4:

**Figure supplement 1.** MARF1 RRM1 is required to silence mRNAs containing 3'UTRs of MARF1 target mRNAs.

## EDC4-MARF1 interaction inhibits MARF1 binding to target mRNAs

Notwithstanding that EDC4 impairs the ability of MARF1 to silence reporters with MARF-targeted endogenous 3'UTRs, EDC4 had no noticeable impact on the ability of full-length λNHA-MARF1 to repress a RL-5BoxB mRNA (*Figure 5H*). As MARF1 is recruited to these mRNAs via different modes (λN-tag-BoxB tethering as opposed to LOTUS domains) our data suggest that EDC4 does not impair MARF1 activity but rather prevents MARF1 from binding to endogenous target 3'UTRs. To test this hypothesis, we co-transfected RL-MAML1 and FL plasmids into HEK293 cells that express FLAG-tagged MARF1$^{\Delta NYN}$, together with or without a plasmid coding for V5-tagged EDC4. Transfected cells were subsequently UV crosslinked and cell extracts were immunopurified with FLAG antibody. RT-qPCR of immunopurified RNAs demonstrated enrichment of RL-MAML1 mRNA with FLAG-tagged MARF1$^{\Delta NYN}$ (~9 fold) (*Figure 6D*). In contrast, the ability of MARF1 to co-precipitate RL-MAML1 mRNA in EDC4 overexpressing cells was dramatically impaired (~2 fold). These data therefore support the hypothesis that EDC4 impairs MARF1 repression of endogenous target mRNAs by preventing MARF1 from binding to targeted 3'UTRs.

## Discussion

In this study we identify endogenous targets of the human endonuclease MARF1. We show that MARF1 predominantly binds the 3'UTRs of target mRNAs and reveal that different RNA binding domains in MARF1 are required for initial target binding as well as subsequent target decay. Importantly, we uncover a novel role for the enhancer of mRNA decapping protein EDC4 in that its binding to MARF1 localizes MARF1 to P-bodies and impairs MARF1-mediated repression.

## MARF1 binds target mRNAs 3'UTRs via core LOTUS repeats

Mammalian MARF1 is an endoribonuclease that possesses multiple RNA binding domains and physically interacts with the mRNA decapping machinery (*Bloch et al., 2014*; *Nishimura et al., 2018*). Knocking out MARF1 or impairing its RNase activity in female mice leads to meiotic arrest during oogenesis and to wide-spread changes to the transcriptome (*Su et al., 2012a*; *Yao et al., 2018*). Our iCLIP analysis of MARF1 in HEK293 cells identified 108 mRNAs bound by MARF1 and demonstrates that MARF1 binds to the majority of these mRNAs via their 3'UTRs. Moreover, we provide evidence using mRNA reporters containing 3'UTRs of endogenous target mRNAs that MARF1 represses them in a NYN-dependent manner. While luciferase reporter assays demonstrated that MARF1 represses gene expression in a NYN-dependent manner, some degree residual repression of the reporters was observed in cells expressing a MARF1 variant lacking its endonuclease NYN domain (*Figure 2D*). It is possible that MARF1$^{\Delta NYN}$ binding to select mRNA 3'UTRs may alter 3'UTR topology in such a way as to impair reporter mRNA translation without altering their stability. This is further supported by the fact that MARF1$^{\Delta NYN}$ did not significantly repress mRNA levels of several target mRNAs as compared to LacZ (*Figure 2E and G*).

Importantly, many MARF1-targeted mRNAs identified by iCLIP are upregulated in MARF1 knockout mouse oocytes as compared to wild-type oocytes. Notwithstanding the validation of a number of MARF1-targeted mRNAs, prediction algorithms including DREME (*Bailey, 2011*) did not identify a motif within the iCLIP peaks, suggesting that MARF1 may recognize structural motifs rather than specific nucleotide sequences. Our CLIP experiments were carried out in the presence of endogenous EDC4, which we now know is sufficient to partially inhibit MARF1 from interfacing with target mRNAs in a dose-dependent manner (*Figure 5*). It is possible that MARF1-targeted mRNAs are underrepresented in our CLIP dataset due to the presence of endogenous EDC4. It will therefore be important to determine if altering EDC4-MARF1 dynamics identifies a larger set of MARF1 target mRNAs.

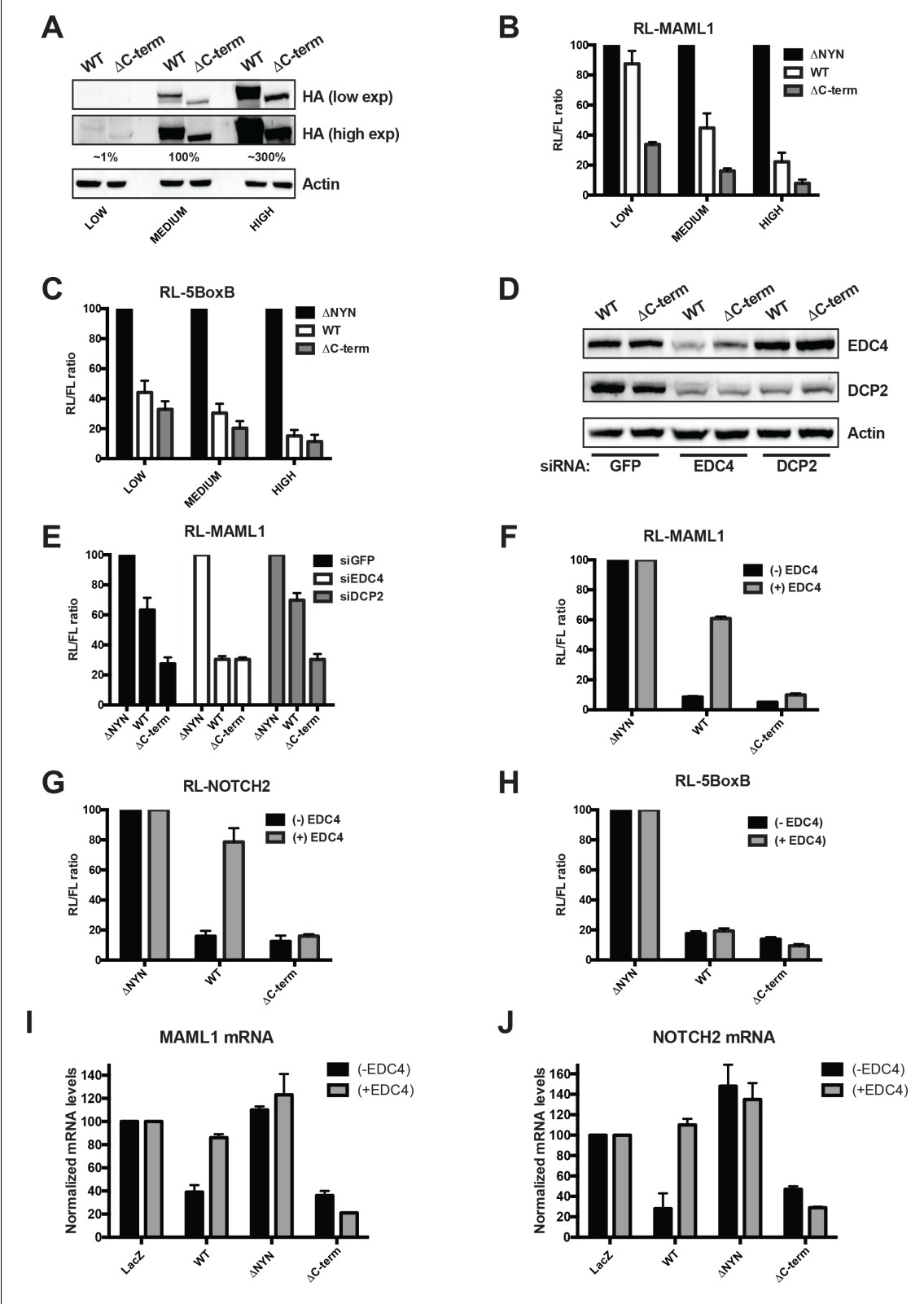

**Figure 5.** EDC4 impairs MARF1 silencing of endogenous target mRNAs. (**A**) Western blot analysis HEK293 cells transfected with different amounts [low (0.1 µg), medium (1.0 µg) and high (3.0 µg)] of wild-type λNHA-MARF1 and λNHA-MARF1$^{\Delta C\text{-term}}$ plasmids. HA Western blot signals are quantified relative to actin and marked below each lane, with 'medium' HA signal set to 100%. (**B and C**) RL activity detected in extracts from HEK293 cells. Cells were cotransfected with constructs expressing the RL-MAML1 reporter (**B**) or RL-5BoxB reporter (**C**), along with FL and different amounts (low, medium

*Figure 5 continued on next page*

*Figure 5 continued*

and high) of wild-type λNHA-MARF1 and λNHA-MARF1$^{\Delta C\text{-term}}$ plasmids. (**D**) Western blot analysis HEK293 cells depleted of EDC4 or DCP2 by siRNA-mediated knockdown. siGFP represents a negative control. (**E**) siRNA-treated cells were subsequently cotransfected with constructs expressing the RL-MAML1 reporter, FL, and low levels of λNHA-MARF1 or λNHA-MARF1$^{\Delta C\text{-term}}$ plasmids, and RL activity detected in extracts. (**F** through **J**) RL activity and mRNA levels detected in extracts from HEK293 cells normalized to FL activity and mRNA levels. Cells were cotransfected with constructs expressing the RL-MAML1 reporter (**F** and **I**), RL-NOTCH2 reporter (**G** and **J**) or RL-5BoxB reporter (**H**), along with FL, V5-tagged EDC4 and high amounts of wild-type λNHA-MARF1 and λNHA-MARF1$^{\Delta C\text{-term}}$ plasmids. All Histograms represented normalized mean values of RL activity (**F** through **H**) or mRNA levels (**G** and **I**) from a minimum of three experiments. RL activity values seen in the presence of λNHA-MARF1$^{\Delta NYN}$ and mRNA levels seen in the presence of λNHA-LacZ were set as 100.

The online version of this article includes the following figure supplement(s) for figure 5:

**Figure supplement 1.** MARF1 C-terminus is sufficient to localize with EDC4 in P-bodies.

Our data indicate that MARF1 is recruited to target mRNAs via its tandem LOTUS domains. The LOTUS domain is a wHTH fold that is conserved from bacteria to humans (*Callebaut and Mornon, 2010*; *Jeske et al., 2017*; *Yao et al., 2018*). The fact that wHTH folds have also been reported to bind double-stranded RNA—including in human La, mouse MARF1 and the signal recognition particle (*Dong et al., 2004*; *Keenan et al., 1998*; *Yao et al., 2018*)—supports a model whereby MARF1 contacts structural elements within the 3'UTRs of target mRNAs. MARF1 proteins contain several (between six and eight) tandem LOTUS domains. *Drosophila* MARF1 protein lacks a NYN domain; however, its first LOTUS domain has been reported to recruit the CCR4-NOT complex to bring about deadenylation of targeted mRNAs (*Zhu et al., 2018*). Both mouse and human MARF1 proteins contain eight LOTUS domains. However, in contrast to results obtained with *Drosophila* MARF1, human MARF1 interacts with the mRNA decapping machinery but not with the CCR4-NOT complex (*Nishimura et al., 2018*). Moreover, tethering an N-terminal fragment of human MARF1 that contains the NYN domain and RRMs but lacks all eight LOTUS domains efficiently represses a reporter mRNA when artificially tethered to it (RL-5BoxB). Although mammalian MARF1 proteins contain eight LOTUS domains, our deletion analyses suggest that certain core LOTUS domains (3 through 5) are critical for MARF1-mediated repression, whereas other adjacent LOTUS domains are not. Importantly, while all MARF1 proteins contain between 6 and 8 LOTUS domains, LOTUS domains 3 and 5 are not only highly conserved across MARF1 orthologs but also share an extremely high degree of homology to each other. While our data does not exclude the possibility that adjacent LOTUS domains also bind to target mRNAs, they do suggest that the central LOTUS domains are critical for the initial steps of target mRNA recognition. Exactly how LOTUS domains 3 and 5 contact MARF1 target elements remains to be explored.

## RRM1 is a NYN coactivator module

All MARF1 proteins that contain a NYN domain (with *Drosophila* MARF1 being the exception) also contain two RRMs, the functions of which are not known. We show here that human MARF1 requires RRM1 to efficiently repress target mRNAs. MARF1 mutants lacking RRM1 or containing amino acid substitutions within its putative RNA binding surface are unable to efficiently silence reporter mRNAs containing MARF1-targeted 3'UTRs. Importantly, these mutants were also unable to silence the RL-5BoxB reporter when artificially tethered to it. In marked contrast to this, a MARF1 deletion mutant lacking its LOTUS domains, which could repress the RL-5BoxB construct was unable to silence reporters with endogenous 3'UTR sequences targeted by MARF1. All in all, these data suggest that RRM1 does not play a role in initial target mRNA recognition but rather in subsequent NYN-mediated cleavage of a targeted mRNA. We speculate that RRM1 may function to properly position the NYN domain on a target mRNA in order to promote efficient cleavage. The fact that all MARF1 proteins that contain a NYN domain also maintain RRM1, whereas the *Drosophila* MARF1 protein that lacks a NYN domain does not contain an analogous domain to RRM1, lends credence to this model.

## A non-canonical role for EDC4 in regulating mRNA decay

The mRNA decapping machinery, comprising DCP1 and DCP2, EDC4 and other factors, plays an important role in promoting mRNA turnover by hydrolyzing the mRNA 5' cap, thus making it

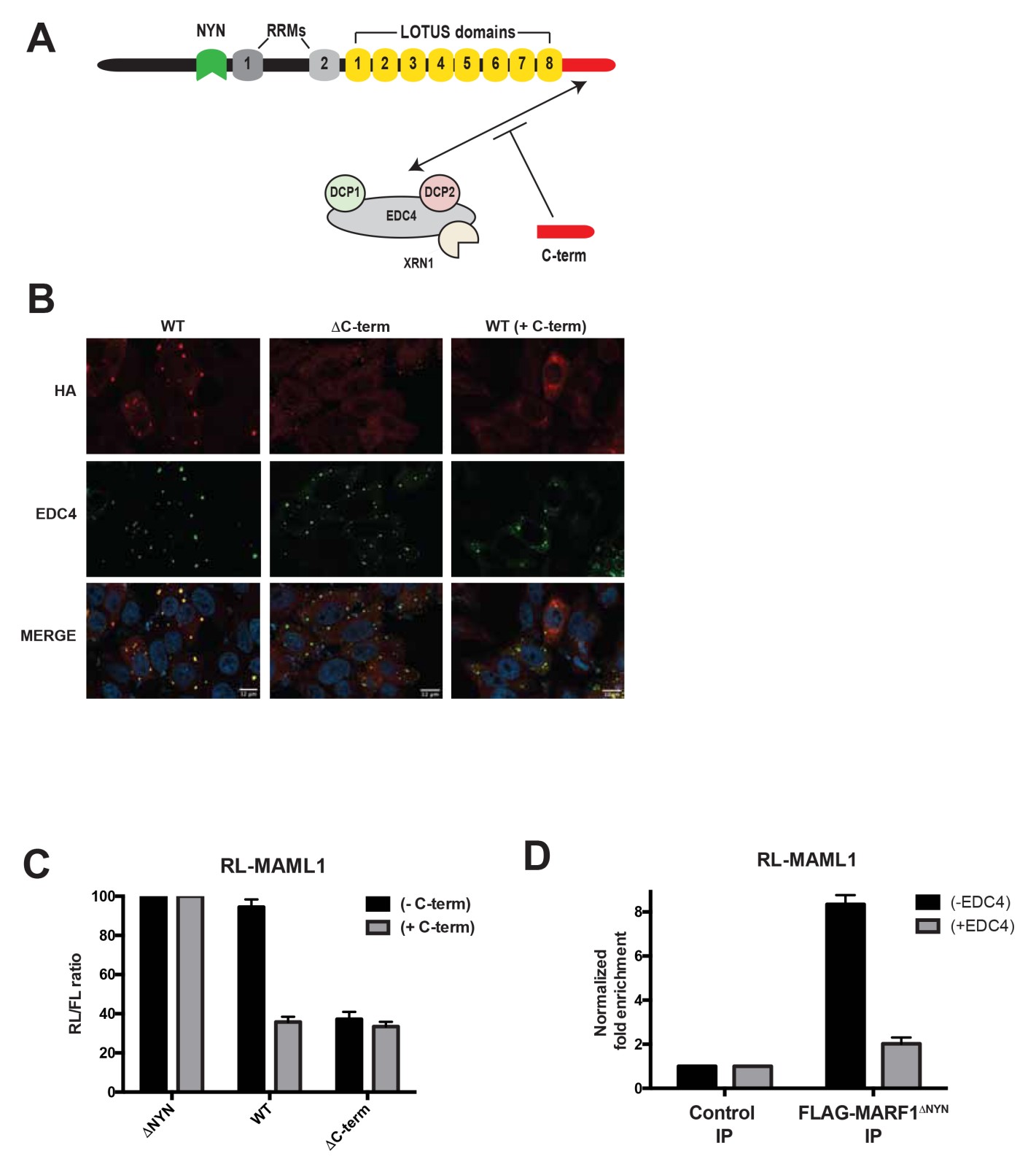

**Figure 6.** EDC4-MARF1 interaction localizes MARF to P-bodies and impairs MARF1 silencing. (**A**) Schematic model of MARF1 C-terminus competition assay. MARF1 contains a C-terminal motif (red) that interacts with the mRNA decapping machinery, including EDC4, DCP1, DCP2 and XRN1. Co-transfecting a plasmid encoding the MARF1 C-terminal motif may compete with full-length MARF1 in binding to the decapping machinery. (**B**) Confocal fluorescence micrographs of fixed HeLa cells expressing wild-type λNHA-MARF1 (with or without FLAG-tagged MARF1[C-term]) and λNHA-MARF1[ΔC-term],

*Figure 6 continued on next page*

*Figure 6 continued*

along with EDC4. Cells were stained with anti-HA (red) and anti-EDC4 (green) antibodies. The merged images show the HA signal in red and the EDC4 signal in green. (C) RL activity detected in extracts from HEK293 cells expressing the indicated proteins. Cells were co-transfected with constructs expressing the RL-MAML1 reporter, FL, low amounts of indicated λNHA-MARF1 constructs, along with/without a plasmid coding for MARF1 C-terminal motif. Histograms represented normalized mean values of RL activity from a minimum of three experiments. RL activity values seen in the presence of λNHA-MARF1$^{\Delta NYN}$ were set as 100. (D) RNA immunoprecipitation of RL-MAML1 reporter by FLAG-tagged MARF1$^{\Delta NYN}$ in HEK293 cells plus/minus EDC4 overexpression.

The online version of this article includes the following figure supplement(s) for figure 6:

**Figure supplement 1.** MARF1 C-terminus interfaces with the mRNA decapping machinery and localizes to P-bodies.

accessible to 5′−3′ decay by the XRN1 exonuclease. EDC4 also associates with the miRISC and enhances microRNA-mediated mRNA decay (*Brodersen et al., 2008*; *Eulalio et al., 2007*; *Nishihara et al., 2013*). Similar to the miRISC, MARF1 physically interacts with mRNA decapping factors, including EDC4 (*Bloch et al., 2014*; *Nishimura et al., 2018*). Importantly, we show here that instead of stimulating the decay of MARF1-targeted mRNAs, EDC4 has the opposite effect. Overexpressing EDC4 impairs MARF1 repression of target RNAs but has no effect on the silencing potential of a MARF1 mutant that cannot bind EDC4. In addition, depleting endogenous EDC4 or preventing MARF1 association with EDC4 enhances MARF1-mediated repression. Importantly, our data indicate that EDC4 only impairs MARF1 repression only when it is recruited to a target mRNA via its LOTUS domains, whereas EDC4 association with MARF1 has no impact on MARF1 repression when it is artificially tethered to a reporter mRNA in a LOTUS-independent manner (i.e. λN-BoxB tethering). Moreover, RNA immunoprecipitation experiments indicate that EDC4 negatively regulates MARF1-mediated repression by preventing target mRNA association as opposed to inhibiting NYN-mediated target cleavage. Based on these data, we hypothesize that EDC4 binding to the MARF1 C-terminal motif precludes the LOTUS domains from binding to endogenous MARF1 target mRNAs (*Figure 7*). To our knowledge, this is the first example of the mRNA decapping machinery stabilizing mRNAs (albeit indirectly) by inhibiting an RNA binding protein from interacting with its targets, rather than promoting their turnover.

While metazoan cells contain a number of mRNA-targeting endonucleases including SMG6 and Regnase, MARF1 is unique in that it physically interacts with mRNA decapping factors. Moreover,

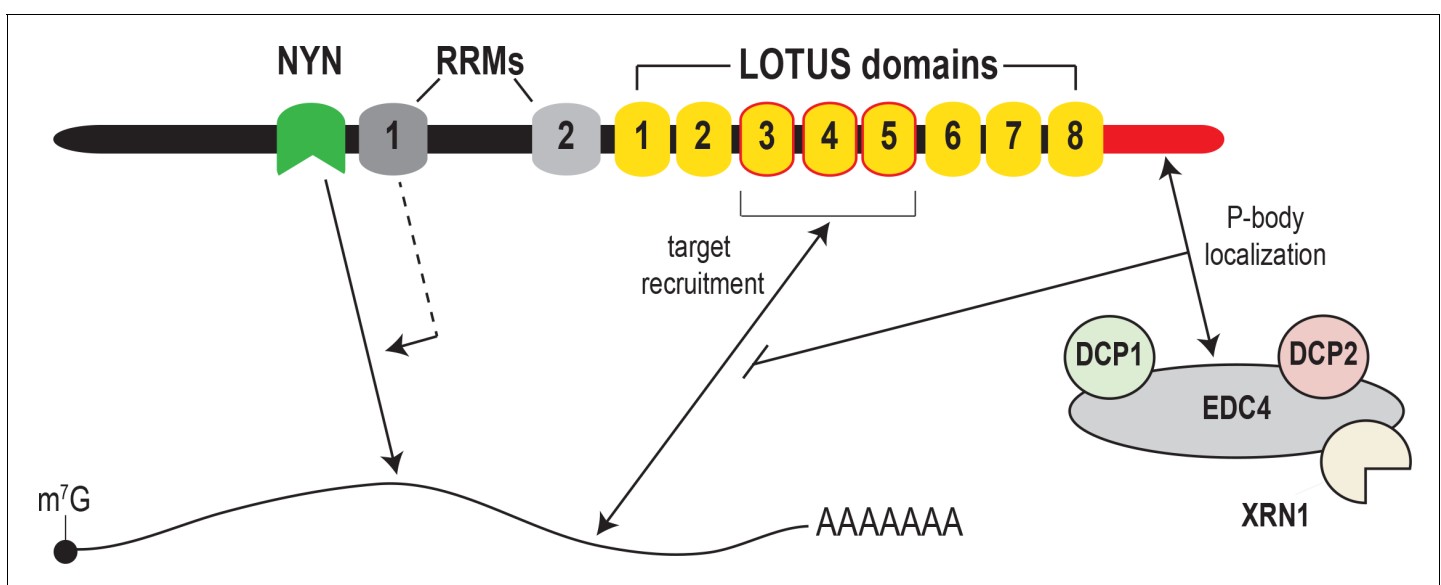

**Figure 7.** Model for MARF1-mediated mRNA decay. MARF1 recognizes target mRNAs via LOTUS domains 3 through 5. Subsequently, RRM1 enhances NYN-mediated cleavage of target mRNAs, potentially by assisting in positioning the NYN domain on target. EDC4 regulates MARF1-mediated repression by interacting with the MARF1 C-terminus, segregating MARF1 to P-bodies and preventing MARF1 from binding target mRNAs, potentially by interfering with LOTUS domain-RNA interactions.

SMG6 does not require decapping proteins to promote nonsense-mediated mRNA decay (*Boehm et al., 2016*). We speculate that this form of regulation may serve to prevent MARF1 protein from binding to mRNAs when MARF1 is expressed at low levels. Only when MARF1 gene expression is dramatically upregulated, as is the case during certain stages of oogenesis (*Su et al., 2012a*; *Su et al., 2012b*) and bind to target mRNAs in a LOTUS domain-dependent manner. Whether EDC4 has a similar mode of action for other RNA binding proteins remains to be investigated.

EDC4 is a core component of P-bodies, membraneless granules that contain proteins involved in mRNA metabolism (*Standart and Weil, 2018*; *Yu et al., 2005*). However, instead of promoting mRNA decay, mRNAs trapped in P-bodies exist in a translationally-repressed state (*Hubstenberger et al., 2017*). Our data demonstrate that the MARF1 endonuclease localizes to P-bodies via its interaction with EDC4. However, it also suggests that MARF1 is 'trapped' in P-bodies and unable to interface with endogenous target mRNAs. Taken together, these data suggest that P-bodies may also regulate mRNA metabolism by impounding specific effectors of mRNA decay (e.g. MARF1) such that they are unable to efficiently bind target mRNAs.

# Materials and methods

## Key resources table

| Reagent type (species) or resource | Designation | Source or reference | Identifiers | Additional information |
|---|---|---|---|---|
| Gene (*Homo sapiens*) | MARF1 | | HGNC: HGNC:29562 | |
| Gene (*Homo sapiens*) | EDC4 | | HGNC: HGNC:17157 | |
| Cell line (*Homo sapiens*) | 293T | ATCC | CRL-3216 | Cell line maintained in DMEM + 10% FBS, 50 U/mL penicillin, and 50 ug/mL streptomycin |
| Cell line (*Homo sapiens*) | HeLa | ATCC | CCL-2 | Cell line maintained in DMEM + 10% FBS, 50 U/mL penicillin, and 50 ug/mL streptomycin |
| Cell line (*Homo sapiens*) | Flp-In T-REx 293 | Thermo Fisher | R78007 | Cell line maintained in DMEM + 10% FBS, 50 U/mL penicillin, and 50 ug/mL streptomycin |
| Antibody | Anti-HA mouse monoclonal | Covance | 901533 | WB(1:1000); IF(1:200) |
| Antibody | Anti-FLAG M2 mouse monoclonal | Sigma-Aldrich | F1804 | WB(1:1000); IF(1:500) |
| Antibody | Anti-V5 Rabbit monoclonal | Cell Signaling | 13202 | WB(1:1000); IF(1:500) |
| Antibody | Anti-human/mouse EDC4 Rabbit monoclonal | Bethyl | A300-745A | WB(1:1000); IF(1:500) |
| Antibody | Anti-human/mouse DCP2 Rabbit monoclonal | Bethyl | A302-597A | WB(1:1000) |
| Antibody | Anti-human Actin Mouse monoclonal | Cell Signaling | 3700 | WB(1:30000) |
| Antibody | Alexa Fluor 488 anti-rabbit | Invitrogen | A32731 | IF(1:500) |
| Antibody | Alexa Fluor 594 anti-mouse | Invitrogen | A32742 | IF(1:500) |
| Recombinant DNA reagent | PLKO.1-Puro (plasmid) | Sigma-Aldrich | RRID :Addgene10878 | shRNA backbone used for selecting transfected cells with puromycin |
| Recombinant DNA reagent | pCI-neo | Promega | E1731 | For all constructs labelled 'LNHA' and 'RL' |

*Continued on next page*

*Continued*

| Reagent type (species) or resource | Designation | Source or reference | Identifiers | Additional information |
|---|---|---|---|---|
| Recombinant DNA reagent | pBABE-3x FLAG-MARF1$^{Cterm}$ | *Nishimura et al., 2018* | | Construct expressing a fragment of MARF1 previously validated by our group to be sufficient to physically interact with the mRNA decapping machinery. |
| Recombinant DNA reagent | pcDNA-DEST40 | Thermo Fisher | 12274015 | For expression of gateway cloned V5-tagged EDC4 |
| Sequence-based reagent | MAML1 3'UTR cloning primers | This paper | PCR Primer | FWD: CGGCGCCGCTCTAGAG GTGTTGGGACAGCAGGATA REV: CGGCGCCGCGCGGCCGC CATAGCTCCCCCAAAACACAC |
| Sequence-based reagent | NOTCH2 3'UTR cloning primers | This paper | PCR Primer | FWD: CGGCGCCGCGTCAGCGA GAGTCCACCTCCAGTGTAGAG REV: CGGCGCCGCGCGGCCGC CATGTTCAAATATCTCACTGAC |
| Sequence-based reagent | ZFP36L2 3'UTR cloning primers | This paper | PCR Primer | FWD: GCAGTAATTCTAGAGGCA AGAGGGCGCCAGTGAGGAGGA REV: GCAGTAATGCGGCCGCCC CAAAAATTTTATTGGGGGAAAAC |
| Sequence-based reagent | ATXN7L3 3'UTR cloning primers | This paper | PCR Primer | FWD: GCAGTAATTCTAGACTTG GGTGCAAGGGATAGCCTTTGG REV: GCAGTAATGCGGCCGCCC AACGGGAGATGCAGTTTATTTAC |
| Sequence-based reagent | FAF2 3'UTR cloning primers | This paper | PCR Primer | FWD: GCAGTAATGCTAGCCCTC CTACCCCAGTCCCTAAAAGAA REV: GCAGTAATGCGGCCGCCT GAAACTCTTTGCTTGGCCTTGGC |
| Sequence-based reagent | CPSF7 3'UTR cloning primers | This paper | PCR Primer | FWD: GCAGTAATTCTAGAGGA GTCTGGTTGGAAGCAAATGTTT REV: GCAGTAATGCGGCCGCTC ACCGACAACAGGGGGACGGGACC |
| Sequence-based reagent | IGF2BP1 3'UTR cloning primers | This paper | PCR Primer | FWD: GCAGTAATGCTAGCGGA GAACAGGCCTGGTGGGAAAGGC REV: GCAGTAATGCGGCCGCGT AGTTACTAGCACTGCTGGTTCCC |
| Sequence-based reagent | PGAM1 3'UTR cloning primers | This paper | PCR Primer | FWD: GGCGCCGCGGCTAGC CCCACCTGCACATGT CACACTGACCAC REV: GGCGCCGCGCGGCCGC ATACTGATATGGAAAA AGGATTTAGTACAG |
| Sequence-based reagent | MARF1$^{ΔNYN}$ cloning primers | This paper | PCR Primer | FWD: GTGCTAGAAAACTTACC CTTCATTTCCGACTTG CCCCCCAGGTTACCAC REV: GGCAAGTCGGAAATGAAG GGTAAGTTTTCTAGCA CCTGTCCAGCTACTGC |
| Sequence-based reagent | MARF1$^{ΔRRM1}$ cloning primers | This paper | PCR Primer | FWD: AAAAATGCCACAGACTC CAAAAAATAGAGAACTCTGTG REV: TTCTCTATTTTTTGGAGTCT GTGGCATTTTTAGTGGTAACCTG |
| Sequence-based reagent | MARF1$^{ΔRRM2}$ cloning primers | This paper | PCR Primer | FWD: TGCCCAGACCCACTCTCTT TACTGAGTGCAGAAACAATG REV: CACTCAGTAAAGAGAGTGG GTCTGGGCAGTCGGCTTCGCTG |
| Sequence-based reagent | MARF1$^{Y513A/Y515A}$ cloning primers | This paper | PCR Primer | FWD: GCTCGCTGTTGCTAACCT ACCAGCAAATAAGGATGGC REV: GTAGGTTAGCAACAGCGAG CAGAGTGTGGCACTGTGGC |

*Continued on next page*

*Continued*

| Reagent type (species) or resource | Designation | Source or reference | Identifiers | Additional information |
|---|---|---|---|---|
| Sequence-based reagent | MARF1$^{I552A}$ cloning primers | This paper | PCR Primer | FWD: CTGCAGTGCAGCTCTCCGC TTCATAAACCAAGATAGTG REV: GAAGCGGAGAGCTGCACTG CAGCCTGTGATACTCAGCAC |
| Sequence-based reagent | MARF1$^{F582A}$ cloning primers | This paper | PCR Primer | FWD: TTGTGTCAGCTACTCCAA AAAATAGAGAACTCTGTGAAAC REV: TTTGGAGTAGCTGACAC AATGATCCTATTACCAAAGACATC |
| Sequence-based reagent | MARF1$^{\Delta LOTUS1}$ cloning primers | This paper | PCR Primer | FWD: TGGTCTCACTTGCCACCGGGG CTGCCAGCAAATCACTACCCA GCAGTCAGGCCCGCCAGA REV: GGGGCTCTGGCGGGCCT GACTGCTGGGTAGTGATTTGC TGGCAGCCCCGGTGGCA |
| Sequence-based reagent | MARF1$^{\Delta LOTUS2}$ cloning primers | This paper | PCR Primer | FWD: TCTTACAAGATTCCTTT TGTGATTCTTTCTATTCA CAACAAGCCCCCGCC REV: AGTGTTGGGAGGCGG GGGCTTGTTGTGAATAGA AAGAATCACAAAAGGAA |
| Sequence-based reagent | MARF1$^{\Delta LOTUS3}$ cloning primers | This paper | PCR Primer | FWD: CGTTCGAAGAGTCCTGTAGG TAACCCCCAGCACAGGGC CCAGGTGAAGCGCTTTA REV: CTGAGTAAAGCGCTTCACCT GGGCCCTGTGCTGGGGGT TACCTACAGGACTCTTC |
| Sequence-based reagent | MARF1$^{\Delta LOTUS4}$ cloning primers | This paper | PCR Primer | FWD: CGTCTGCTGACCCTTACCCA CAGGGCCCAGCCCAAAAGA GAACGCACTCAGGATG REV: TATTTCATCCTGAGTGCGTTC TCTTTTGGGCTGGGCCCTG TGGGTAAGGGTCAGC |
| Sequence-based reagent | MARF1$^{\Delta LOTUS5}$ cloning primers | This paper | PCR Primer | FWD: AGAGAACGCACTCAGGATGA AATAGAAAGGCTTTTCTTC GAGCGGTTCAAAGCTC REV: AGCTAGAGCTTTGAACCGCTC GAAGAAAAGCCTTTCTATTTCA TCCTGAGTGCGT |
| Sequence-based reagent | MARF1$^{\Delta LOTUS6}$ cloning primers | This paper | PCR Primer | FWD: TGTCAGAGTAAGGATCTTTT CTTCGAGCGGATCAACCGAAA GTCTCTGCGATCTC REV: AGTGAGAGATCGCAGAGACTT TCGGTTGATCCGCTCGAAGAA AAGATCCTTACTC |
| Sequence-based reagent | MARF1$^{\Delta LOTUS7}$ cloning primers | This paper | PCR Primer | FWD: AGACAGATTCAGCTGATCAAC CGAAAGTCTACAAGTCTGTATTTGTTTGC REV: CACATTCTTTGCAAACAAATAC AGACTTGTAGACTTTCGGTTGATCAGCT |
| Sequence-based reagent | Full length MARF1 cloning primers | This paper | PCR Primer | FWD: GGACGATCTGCAATTGGAAG GAAACGGAACTGAGAACTCCTGC REV: GCGCCGCGCGGCCGCTTAAAG CTTGGTTATAGGTGCTAAGGAAAAG |
| Sequence-based reagent | MARF1$^{\Delta Cterm}$ cloning primers | This paper | PCR Primer | FWD: GGACGATCTGCAATTGGAA GGAAACGGAACTGAGAACTCCTGC REV: GCGCCGCGCGGCCGCTTA GAGACTGAGTGAACTCAAACGAC |
| Sequence-based reagent | MARF1$^{N-term}$ cloning primers | This paper | PCR Primer | FWD: GGACGATCTGCAATTGGA AGGAAACGGAACTGAGAACTCCTGC REV: GCGCCGCGCGGCCGC TTACCCGGTGGCAAGTGAGACCAGG |

*Continued on next page*

*Continued*

| Reagent type (species) or resource | Designation | Source or reference | Identifiers | Additional information |
|---|---|---|---|---|
| Sequence-based reagent | EDC4 Gateway cloning primers | This paper | PCR Primer | FWD: GGGGACAAGTTTGTACAAAAAAGCAGGCTACCATGGCCTCCTGCGCGAGCATCGACATCG REV: GGGGACCACTTTGTACAAGAAAGCTGGGTCAGGGAGGCTGGGGGTCACGA |
| Sequence-based reagent | FL qPCR primers | This paper | | FWD: CCTTCGATAGGGACAAGACAA REV: AATCTCACGCAGGCAGTTCT |
| Sequence-based reagent | RL qPCR primers | This paper | | FWD: GAGTTCGCTGCCTACCTGGAGCCAT REV: GGATCTCGCGAGGCCAGGAGAG |
| S1equence-based reagent | GAPDH qPCR primers | This paper | | FWD: GTGGAGATTGTTGCCATCAACGA REV: CCCATTCTCGGCCTTGACTGT |
| Sequence-based reagent | MAML1 qPCR primers | This paper | | FWD: GACTCTCTCAACAAAAAGCGTCT REV: AGGAAATGACTCACTGGGGTTA |
| Sequence-based reagent | NOTCH2 qPCR primers | This paper | | FWD: CTCCAGGAGAGGTGTGCTTG REV: TGATGTCTCCCTCACAACGC |
| Sequence-based reagent | GFP siRNA | Dharmacon | D-001940-01-05 | Accell eGFP control siRNA |
| Sequence-based reagent | EDC4 siRNA | Dharmacon | L-016635-00-0005 | SMARTpool |
| Sequence-based reagent | DCP2 siRNA | Dharmacon | L-008425-01-0005 | SMARTpool |
| Peptide, recombinant protein | Actinomycin D | Sigma-Aldrich | A1410 | |
| Commercial assay or kit | GoTaq qPCR Master Mix | Promega | A6001 | Reagent for all qPCR assays |
| Commercial assay or kit | Dual-Luciferase Assay | Promega | E1910 | Reagent for all luciferase assays |
| Other | DAPI stain | Invitrogen | D1306 | (1 μg/mL) |

## Cell culture and transfection

Human embryonic kidney HEK293 cells and epithelioid carcinoma HeLa cells were obtained from ATCC. Flp-In T-REx 293 Cell Line were obtained from Thermo Fisher Scientific. Cell lines identity was established via morphology but has not been authenticated and all cells were tested negative for mycoplasma contamination. HEK293 and HeLa cells were grown in Dulbecco's modified Eagle's medium (DMEM) supplemented with 10% fetal bovine serum, 50 U/mL penicillin, and 50 ug/mL streptomycin. Plasmid transfections were performed using polyethyleneimine (PEI).

## iCLIP experimental procedures

Crosslink immunoprecipitation experiments were conducted using a modified single-end (se)CLIP approach (*Van Nostrand et al., 2017*). All experiments were conducted using HEK293 cells (negative control), or HEK293 cells that express FLAG-tagged MARF$^{\Delta NYN}$. Briefly, FLAG-tagged MARF1$^{\Delta NYN}$ expression was induced in a 15 cm dish of HEK293 cells with 100 ng/mL of doxycycline for 12 hr. Following induction, $2 \times 10^7$ cells were lysed in iCLIP lysis buffer and sonicated (BioRuptor). Lysates were treated with RNAseI (Thermo) to shear RNA, after which FLAG-tagged MARF1$^{\Delta NYN}$-RNA complexes were immunoprecipitated with FLAG antibody. Stringent washes were performed, during wish RNA was dephosphorylated with FastAP (Thermo) and T4 PNK (NEB). Following dephosphorylation, a 3' RNA adaptor was ligated onto the RNA with T4 RNA ligase (NEB). Protein-RNA complexes were resolved by SDS-PAGE, transferred to nitrocellulose membranes, and RNA was isolated from the membrane region corresponding to the migration of MARF1 and 75 kDa above. Isolated RNA was subsequently reverse transcribed with AffinityScript (Agilent) and the 3' DNA adaptor was ligated onto the cDNA with T4 RNA ligase (NEB). Libraries were then amplified with Q5 PCR mix (NEB).

## seCLIP-seq read processing and clustal analysis

Biological duplicate libraries were sequenced by single-end 50 bp sequencing on a Hi-Seq2500 (Sick Kids, Toronto). RNA sequencing reads were trimmed using Trimmomatic (v0.32) (*Bolger et al., 2014*), removing adaptor and other Illumina-specific sequences, and low-quality bases at the end of each read, using a 4 bp sliding window to trim where average window quality fell below 30 (phred33 <30). Trimmed reads with less than 15 bases were discarded. The resulting clean set of reads were then aligned to the hg19 (GRCh37) reference genome using STAR (v2.3.0e) (*Dobin et al., 2013*) with default parameters. Reads mapping to more than 10 locations in the genome (MAPQ <1) were discarded. Secondary alignment reads were removed using samtools (v1.9) (*1000 Genome Project Data Processing Subgroup et al., 2009*) and duplicates were removed using Picard (v2.10.7) MarkDuplicates. Peaks were then called on the resulting bam files using macs2 (v2.1.1) callpeak using the nomodel option, an extension size of 100 and a FDR threshold of 0.05. The list of final peaks reported is the intersection of peaks detected in the pairwise peak calling of MARF1 replicates (n = 2) against control without flag replicates (n = 2). Peaks were then annotated with Homer (v4.10) annotatePeaks using the hg19 reference. Integrative Genomic Viewer (IGV) was used for visualization (*Thorvaldsdóttir et al., 2013*).

## Plasmids

Plasmids pCI-λNHA-MARF1 variant plasmids were generated by conventional molecular cloning techniques using MfeI and NotI restriction enzyme sites. Point and deletion mutations within the coding sequence of MARF1 were made through site-directed mutagenesis with Phusion Hot-Start II DNA polymerase. Similarly, *Renilla* luciferase (RL) reporter plasmids containing gene-specific 3'UTR regions were generated through conventional cloning using XbaI and NotI restriction enzyme sites. The RL-5BoxB and firefly luciferase (FL) plasmids have been previously described (*Nishimura et al., 2018*). V5-tagged EDC4 was generated by Gateway cloning pDONR-EDC4 into pcDNA-DEST40 vector (thermo).

## siRNAs and antibodies

Antibodies against HA, FLAG, V5, and 4E-T were purchased from Covance, Sigma, Cell Signaling, and Abcam, respectively. The antibodies against EDC4 and DCP2 were both purchased from Bethyl Laboratories. For siRNA-mediated knockdowns HEK293T cells were seeded at a density of ~15% confluency and transfected with siRNAs against GFP, EDC4 and DCP2 (Dharmacon) at a final concentration of 150 nM using Lipofectamine 2000 reagent (Invitrogen). 24 hr post-transfection, cell culture media was replaced and cells were permitted to grow for an additional 48 hr before being harvested.

## Luciferase and RT-qPCR analyses

HEK293T cells were seeded at a density of ~35% confluency and transfected 24 hr post-seeding. After an additional 24 hr post-transfection, cells were harvested and lysed in Passive Lysis Buffer (Promega). The activity levels of the *Renilla* (RL) and firefly (FL) luciferase reporters was measured using Dual-Luciferase Assay (Promega). Cell lysate was also analyzed by western blotting in order to determine relative protein expression levels. For RT-qPCR analysis HEK293T cells were seeded at a density of ~20% confluency and co-transfected with MARF1 variants, RL-MAML1, FL (control) and an empty puromycin-resistance selection cassette 24 hr post-seeding. 24 hr post-transfection, the cell culture media was replaced with complete cell culture media containing 2 ug/mL of puromycin. After an additional 24 hr post-puromycin selection, cells were harvested and RNA was extracted using EZ-10 Spin Column RNA Miniprep Kit (Biobasic). Extracted RNA was DNaseI treated (Invitrogen) for 1 hr and inactivated using inactivating reagent (Invitrogen). The DNaseI treated RNA was random hexamer-primed and reverse transcribed following the protocol for Maxima H Minus Reverse Transcriptase (Thermofisher). Quantitative PCR on the generated cDNA was carried out with GoTaq qPCR Master Mix (Promega) and primers (IDT) against the ORF of RL-MAML1 and FL (control) plasmids and against the coding sequence of endogenous GAPDH (control), MAML1, and NOTCH2 transcripts.

## Structural modelling of RRM1

The structural model of the MARF1 RRM1 domain was generate using the Phyre2 structure prediction server (*Kelley et al., 2015*). Amino acids putatively involved in RNA binding were identified by superposition with structures of other RRM domain proteins, including Nab3 (*Hobor et al., 2011*).

## Immunofluorescent staining

HeLa cells were seeded at a confluency of ~25% grown on coverslips for 24 hr. Post-seeding, cells were transfected with combinations of HA-tagged MARF1$^{WT}$ or MARF1$^{\Delta C-term}$, V5-tagged EDC4 and 3xFLAG-tagged MARF1$^{C-term}$. 24 hr post-transfection, cells were washed twice with PBS and then subsequently fixed with 10% formaldehyde in PBS for 20 min. After formaldehyde fixing, cells were washed four times with PBS and then permeabilized with 0.1% triton in PBS (PBS-T) for 20 min. After permeabilization, the samples were blocked in 4% bovine serum albumin (BSA) in PBS at room temperature (RT) for 1 hr. Primary antibodies were diluted 1:200 for anti-HA and 1:500 for anti-FLAG and anti-EDC4 in 4% BSA. Diluted antibodies were added to the coverslips after blocking and left to incubate at 4°C overnight. The next day, cells were washed three times with PBS-t. Secondary antibodies were diluted 1:500 for both Alexa Fluor 488 goat anti-rabbit and Alexa Fluor 594 goat anti-mouse in 4% BSA (ThermoFisher). Secondary antibody was added to the coverslips and incubated at RT for 45 min, shielded from light. Post-secondary incubation, coverslips were washed once with PBS-t and then three additional times with PBS to remove any residual triton. Nuclei were stained with DAPI for 15 min at RT, shielded from light. Coverslips were washed four times with PBS before being mounted onto glass slides with ProLong Gold media (ThermoFisher). Images were taken using a Zeiss Confocal LSM 800 microscope at 40X magnification and processed with Fiji to add scale bars.

## RNA immunoprecipitation assay

Experiments were conducted using HEK293 cells that express FLAG-tagged MARF$^{\Delta NYN}$. Briefly, FLAG-tagged MARF$^{\Delta NYN}$ expression was induced in a 10 cm dish of HEK293 cells using 2 ug/mL of doxycycline for 24 hr. Concurrently, cells were transfected with either plasmids coding for GFP (control) or V5-tagged EDC4. 24 hr post-transfection, cells were washed with PBS and UV crosslinked at 150 mJ cm$^{-2}$. Cells were harvested and lysed in lysis buffer (50 mM Tris-HCl, pH 7.4; 100 mM NaCl; 1% NP40 (Igepal CA630); 0.1% SDS; 0.5% sodium deoxycholate; protease inhibitor; in RNase-free water). Lysates were incubated overnight with empty protein G magnetic beads (control) or beads coupled to FLAG antibody. Aliquots were taken for RNA and Protein inputs. Beads were then magnetized and stringently washed as outlined in the eCLIP protocol. 20% of the washed beads were taken for validation by western blot with the 'Protein Input' samples. The remaining beads were Proteinase K treated for 20 min at 37C, shaking at 1200 rpm. Post-proteinase K treatment, input and IP samples were RNA-extracted using EZ-10 Spin Column RNA Miniprep Kit (Biobasic). Extracted RNA was DNaseI-treated (Invitrogen) for 1 hr and inactivated using inactivating reagent (Invitrogen). RNA was reverse transcribed using random hexamer following the protocol for Maxima H Minus Reverse Transcriptase (Thermofisher). Quantitative PCR on the generated cDNAs was carried out with GoTaq qPCR Master Mix (Promega) and primers (IDT) against the ORF of our RL-MAML1 and FL (control) plasmids. To measure MARF1-target mRNA enrichment levels, the input and IP samples were first analyzed separately. The CT values of the RL-MAML1 mRNA as determined by qPCR were first normalized to the FL mRNA CT values ($\Delta$CT). This $\Delta$CT value for the FLAG immunoprecipitation samples was then normalized to the $\Delta$CT value of the uncoupled beads ($\Delta\Delta$CT). Finally, the relative fold-change of the $\Delta\Delta$CT value was determined by calculating $2^{-\Delta\Delta CT}$. After the relative fold-changes were calculated, the enrichment between the IP and input values was determined by dividing the IP relative fold-change by the input relative fold-change.

## mRNA stability assays

HEK293 cells were transiently transfected with plasmids coding for RL-MAML1, FL and various λNHA-expressing proteins using PEI reagent (Polysciences). 24 hr later, cells were treated with Actinomycin D (5 µg/ml) for various amounts of time and lysed. Total RNA was extracted and RT-qPCR

was performed on these samples as outlined above with primers against the coding sequences of RL and FL.

## Acknowledgements

We would like to thank members of the Fabian lab for their input into the project, including Tamiko Nishimura for generating HEK293 cells stably expressing doxycycline-inducible F-MARF1$^{\Delta NYN}$. We would also like to thank Martin Jinek and Tobias Brandmann for their expertise with structural modelling. This work was supported by a Canadian Institutes of Health Research (CIHR) grant (MOP-130425) and a Natural Sciences and Engineering Research Council of Canada (NSERC) Discovery grant (RGPIN-2015–03712) to MRF and Fonds de Recherche du Québec- Santé (FRQS) Chercheur-Boursier Junior two and CIHR New Investigator awards to MRF.

## Additional information

### Funding

| Funder | Grant reference number | Author |
| --- | --- | --- |
| Canadian Institutes of Health Research | MOP-130425 | Marc Fabian |
| Natural Sciences and Engineering Research Council of Canada | RGPIN-2015-03712 | Marc Fabian |
| Fonds de Recherche du Québec - Santé | Chercheurs-boursiers Junior two | Marc R Fabian |
| Canadian Institutes of Health Research | New Investigator award | Marc R Fabian |

The funders had no role in study design, data collection and interpretation, or the decision to submit the work for publication.

### Author contributions

William R Brothers, Conceptualization, Formal analysis, Writing - original draft, Writing - review and editing; Steven Hebert, Data curation, Formal analysis, Writing - review and editing; Claudia L Kleinman, Resources, Formal analysis, Writing - review and editing; Marc R Fabian, Conceptualization, Supervision, Writing - original draft, Writing - review and editing

### Author ORCIDs

William R Brothers (iD) https://orcid.org/0000-0001-8962-7974
Marc R Fabian (iD) https://orcid.org/0000-0003-3700-7604

### Decision letter and Author response

Decision letter https://doi.org/10.7554/eLife.54995.sa1
Author response https://doi.org/10.7554/eLife.54995.sa2

## Additional files

### Supplementary files

• Supplementary file 1. CLIP mapping and cluster statistics.

• Supplementary file 2. List of MARF1 target RNAs identified by iCLIP that overlap with upregulated gene expression in *Marf1*$^{GT/GT}$ and *Marf1*$^{D272A/D272A}$ germinal vesicles (*Yao et al., 2018*).

• Transparent reporting form

### Data availability

Sequencing data have been deposited in GEO under accession code GSE149820.

The following dataset was generated:

| Author(s) | Year | Dataset title | Dataset URL | Database and Identifier |
|---|---|---|---|---|
| Fabian M, Brothers WR, Hebert S, Kleinman C | 2020 | Data from: A non-canonical role for the EDC4 decapping factor in regulating MARF1-mediated mRNA decay | https://www.ncbi.nlm.nih.gov/geo/query/acc.cgi?acc=GSE149820 | NCBI Gene Expression Omnibus, GSE149820 |

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
