## [Decision Letter]

**Acceptance summary:**

The manuscript presents evidence that a recently identified RNA endonuclease MARF1, which was previously shown to be required for normal oogenesis in mice, is negatively regulated by a core component of the mRNA decapping complex, EDC4. In addition, the potential RNA targets of the MARF1 endonuclease are identified by iCLIP, disclosing important mechanistic insights into MARF1 ribonuclease function, its regulation, and its targets in somatic cells.

**Decision letter after peer review:**

Thank you for submitting your article "A non-canonical role for the EDC4 decapping factor in regulating MARF1-mediated mRNA decay" for consideration by *eLife*. Your article has been reviewed by two peer reviewers, and the evaluation has been overseen by a Reviewing Editor and James Manley as the Senior Editor. The reviewers have opted to remain anonymous.

The reviewers have discussed the reviews with one another and the Reviewing Editor has drafted this decision to help you prepare a revised submission.

While the reviewers consider the manuscript to represent a solid and interesting study disclosing important mechanistic insight into the nuclease function of MARF1, they also raise important reservation about the biological relevance. This criticism mostly concerns the uniform use of the luciferase reporter to address mRNA decay relationships as well as the use of somatic cells, where MARF1 has to be overexpressed to allow its study. To consider the manuscript for publication in *eLife*, revision is therefore required addressing the below points.

Essential revisions:

1) According to the model, EDC4 inhibits MARF1 mRNA binding. Since iCLIP was performed in the presence of endogenous EDC4, the low number of identified mRNAs might reflect that over expressed MARF1 is in a rather inactive state. The authors should perform iCLIP with the ΔC-term mutant and compare the results with the full-length dataset (ΔNYN). Alternatively, EDC4 knock down could be performed and MARF1 could be clipped. This would solidify the presented model and add further physiologically relevant data to the study.

2) Nearly all reported effects of MARF1 activity on mRNA decay is provided by reporter constructs with a protein read-out (luciferase activity), and in all cases exogenously expressed MARF1. It is essential to demonstrate, at least initially, that the luciferase output reflects mRNA steady-state levels, and preferably also mRNA decay rates. Also, the demonstrated effects of manipulation with EDC4 expression on luciferase reporters elicited by MARF1 (EDC4 knockdown and EDC4 overexpression; e.g. Figure 5) needs to be recapitulated with endogenous targets.

---

## [Author Response]

Essential revisions:1) According to the model, EDC4 inhibits MARF1 mRNA binding. Since iCLIP was performed in the presence of endogenous EDC4, the low number of identified mRNAs might reflect that over expressed MARF1 is in a rather inactive state. The authors should perform iCLIP with the deltaC-term mutant and compare the results with the full-length dataset (deltaNYN). Alternatively, EDC4 knock down could be performed and MARF1 could be clipped. This would solidify the presented model and add further physiologically relevant data to the study.

We agree with the low number of CLIP-identified mRNAs may be due to EDC4 limiting the number of target mRNAs identified. Doing a CLIP experiment with delta C-term MARF1 mutant that cannot bind to EDC4 would definitely be the experiment to perform. We have modified our Discussion to discuss this important point. However, due to unanticipated circumstances we have not been able to provide this experiment in a timely manner.

2) Nearly all reported effects of MARF1 activity on mRNA decay is provided by reporter constructs with a protein read-out (luciferase activity), and in all cases exogenously expressed MARF1. It is essential to demonstrate, at least initially, that the luciferase output reflects mRNA steady-state levels, and preferably also mRNA decay rates. Also, the demonstrated effects of manipulation with EDC4 expression on luciferase reporters elicited by MARF1 (EDC4 knockdown and EDC4 overexpression; e.g. Figure 5) needs to be recapitulated with endogenous targets.

We had previously demonstrated that luciferase output reflects the steady-state levels of both RL-MAML1 and RL-Notch2 reporter mRNAs in Figure 2D (now Figure 2E). However, our figure labels didn’t clearly reflect this, a mistake that we have now rectified. In addition, we have also assessed the mRNA decay rate for the RL-MAML1 reporter mRNA with actinomycinD pulse experiments (Figure 2F). Our new data demonstrate that wild-type MARF1 destabilizes the RL-MAML1 mRNA, whereas the MARF^ΔNYN^ has no effect on transcript decay rates.